# Effects of Ceragenins and Antimicrobial Peptides on the A549 Cell Line and an In Vitro Co-Culture Model of A549 Cells and *Pseudomonas aeruginosa*

**DOI:** 10.3390/pathogens11091044

**Published:** 2022-09-14

**Authors:** Ozlem Oyardi, Paul B. Savage, Cagla Bozkurt Guzel

**Affiliations:** 1Department of Pharmaceutical Microbiology, Faculty of Pharmacy, Istanbul University, 34116 Istanbul, Turkey; 2Institute of Graduate Studies in Health Sciences, Istanbul University, 34116 Istanbul, Turkey; 3Department of Chemistry and Biochemistry, Brigham Young University, Provo, UT 86001, USA

**Keywords:** ceragenins, antimicrobial peptides, co-culture, biofilm

## Abstract

*Pseudomonas aeruginosa* is an important pathogen that can adhere to host tissues and epithelial surfaces, especially during chronic infections such as cystic fibrosis (CF) lung infections. The effect of ceragenins and antimicrobial peptides (AMP) on this colonization was investigated in a co-culture infection model. After determining the antimicrobial effects of the substances on *P. aeruginosa* planktonic cells, their cytotoxicity on the A549 cell line was also determined. After the A549 cell line was infected with *P. aeruginosa*, the effect of antimicrobials on intracellular bacteria as well as the effects in inhibiting the adhesion of *P. aeruginosa* were investigated. In addition, LDH release from cells was determined by performing an LDH experiment to understand the cytotoxicity of bacterial infection and antimicrobial treatment on cells. CSA-131 was determined as the antimicrobial agent with the highest antimicrobial activity, while the antimicrobial effects of AMPs were found to be much lower than those of ceragenins. The antimicrobial with the lowest IC_50_ value was determined as the combination of CSA-131 with Pluronic F127. CSA-13 has been determined to be the most effective antimicrobial with its effectiveness to both intracellular bacteria and bacterial adhesion. Nevertheless, further safety, efficacy, toxicity, and pharmacological studies of ceragenins are needed to evaluate clinical utility.

## 1. Introduction

*Pseudomonas aeruginosa* is on the priority pathogens list determined by the World Health Organization due to its natural and acquired antimicrobial resistance and virulence factors [1]. Difficulties are encountered in the treatment of *P. aeruginosa* with existing antibiotics due to multi-drug resistance [2,3]. One of the predominant pathogens in the respiratory tract of CF patients is *P. aeruginosa,* and it is a hospital-acquired infection agent [4,5]. Elevated production of mucus in CF lungs facilitates invasion of the lung epithelium by opportunistic pathogens such as *P. aeruginosa*, including adhesion and biofilm formation. *P. aeruginosa* can adhere to living tissues, form biofilms, and also accumulate intracellularly. The presence of intracellular bacteria may cause recurrent infections and makes the treatment of infections, such as those associated with CF, difficult to treat. This is especially true with immunocompromised patients because *P. aeruginosa* may remain in intracellular compartments during chronic respiratory tract infections and is protected from defense mechanisms [6,7]. 

While the spread of antibiotic-resistant bacteria causes mortality and morbidity worldwide, the introduction of new antibiotics to the market is not as rapid. Antibiotic resistance motivates the search for new antimicrobial agents, including development efforts with ceragenins and AMPs. Ceragenins and antimicrobial peptides (AMP) have the potential to be two new classes of antimicrobial agents for the treatment of infections associated with CF, and intensive studies are underway on their efficacy and safety. AMPs are bioactive small molecules that are found in organisms ranging from prokaryotes to humans and take part in the first line of defense against a broad range of pathogens. Some AMPs have been included in clinical studies, and many peptide-based drugs have been approved or are awaiting approval from the Food and Drug Administration (FDA) for the treatment of various diseases [8,9]. Although AMPs display broad-spectrum and immunomodulatory properties, they also have features that limit their clinical use. Some of these are the relatively high cost of producing them on a large scale and their potential to be degraded by proteases released by bacteria [10]. In addition, there are problems and concerns that may be encountered in clinical practice, such as toxicity and immunogenicity, drug resistance, hemolytic activity, and other side effects. In order to overcome such disadvantages, non-peptide synthetic mimics of AMPs have been developed. Ceragenins have been developed as non-peptide AMP mimics, and within the family of ceragenins, multiple examples have been described and characterized as potential therapeutic agents. Ceragenins show lytic activity by destroying the bacterial cell membrane, similar to AMPs [11]. The ubiquity of AMPs and the continued susceptibility of most bacteria to their antibacterial action argue that the generation of stable resistance to these peptides is not likely. However, the complexity of most AMPs and their susceptibility to proteases are some challenges to their clinical use. As non-peptide mimics of AMPs, ceragenins offer comparable mechanisms of bactericidal activity without the accompanying complexity or susceptibility to proteases. The relatively simple structures of ceragenins (Figure 1) are amenable to large-scale production [10].

Drug carrier systems have been developed to reduce the side effects and cytotoxic effects of drugs and to increase drug effectiveness. Pluronics (poloxamers), a drug delivery system, are water-soluble triblock copolymers consisting of polyethylene oxide-polypropylene oxide-polyethylene oxide (PEO-PPO-PEO) and have been approved for pharmaceutical use by FDA. The most well-known member of this family is Pluronic F127 [13]. Pluronic^®^ F-127 is a nonionic drug carrying poloxamer. The Pluronic F127 has been shown to increase the affinity of CSAs to microbial membranes as well as transport them to the targeted tissue [14]. By using Pluronic^®^ F-127, drugs can be retained in micelles and targeted to specific tissues. Encapsulation of CSA-131 in poloxamer micelles decreases cytotoxicity and increases cell membrane selectivity. In this study, the combination of CSA-131 with pluronic F127 (CSA-131P) was also tested for cytotoxicity and effects on co-culture.

It is important to understand the interactions of bacteria with living cells and to test antimicrobials in the co-culture model, as bacteria can typically grow planktonically and in the form of biofilms in the presence of host cells, and may even penetrate cells and accumulate. Therefore, in this study, after evaluating the effects of twelve antimicrobial agents on *P. aeruginosa* and the A549 cell line individually, their simultaneous effects on both human lung cells and *P. aeruginosa* were compared by forming co-culture. 

## 2. Results

### 2.1. The MICs and MBCs

The MIC and MBC values are shown in Table 1. While the MIC values of ceragenins were between 8–32 mg/L, it was found that the MIC values of AMPs were higher than 128 mg/L. Among the ceragenins, CSA-131 (MIC: 4 mg/L) showed activity at the lowest concentration (4 mg/L). MBC values are equal to or twice the MIC values.

### 2.2. MTT Results

Cytotoxicity was determined in an MTT assay. IC_50_ values were calculated using the Graphpad prism program and were shown in Table 2. While the IC_50_ values of ceragenins were in the range of 11.37 ± 3.88 and 40.05 ± 2.65 mg/L, the IC_50_ values of AMPs were determined between 40.76 ± 5.35 and >200 mg/L. The poloxamer form of CSA-131 showed the lowest cytotoxicity among ceragenins (40.05 ± 2.65 mg/L).

### 2.3. Intracellular Activity Results

In studies of the activities of ceragenins and AMPs against intracellular bacteria, CSA-13 showed the highest activity (Figure 2). At a concentration of 5 mg/L of CSA-13, a one-log decrease from the initial bacteria count was observed, while a 10 mg/L concentration caused a three-log decrease, and a 20 mg/L concentration caused a four-log decrease. Similar to CSA-13, the second-generation ceragenin CSA-131, at a concentration of 20 mg/L reduced the initial intracellular bacteria count by three logs. While 20 mg/L concentrations of CSA-90 and CSA-192 among other ceragenins provided one-log reductions, only LL-37, among AMPs, caused as much as a one-log decrease in bacterial counts at 20 mg/L.

### 2.4. Inhibition of Bacterial Adhesion

Bacterial adhesion increased over time in the control group as expected (Figure 3). Adhesion of the bacteria to the cell environment was also confirmed by fluorescent images (Figure 4). The effects of ceragenins and AMPs on the adhesion of *P. aeruginosa* on the A549 cell line were most pronounced with CSA-13; CSA-13 completely prevented *P. aeruginosa* adhesion at 2, 4, and 6 h at a concentration of 20 mg/L. The ability of CSA-13 to inhibit bacterial adhesion was also shown in the fluorescent images in Figure 4. As the concentration of CSA-13 increased, its effectiveness against bacterial adhesion increased. CSA-13 also showed a highly effective inhibitory effect at a concentration of 10 mg/L at which it caused a three-log decrease at each time point, compared to the control group (Figure 3). The effect of CSA-131P on the adhesion of bacteria was found to be more significant than CSA-131. In particular, 20 mg/L CSA-131P caused a reduction of about 2 log at 2 h. Among AMPs, LL-37 was determined as the only AMP that caused a significant decrease. 

### 2.5. LDH Cytotoxity Results

Bacteria alone did not cause significant cytotoxicity in A549 cells at the 2nd, 4th, and 6th h (data not shown). The percentage of LDH release caused by antimicrobials at 2nd, 4th, and 6th h in cell-bacteria co-cultures was determined by comparing the LDH release that occurred in co-cultures without antimicrobials, and the results were shown in Figure 5. According to the results, LDH release from cells increased depending on time and antimicrobial concentrations. Cytotoxicity has not been demonstrated at any concentrations of magainin and cecropin. All the ceragenins tested, except CSA-131P, induced more than 50% cytotoxicity after 6 h at a concentration of 20 mg/L. While CSA-131P did not cause cytotoxicity at 2 h, it showed 7.5% and 33.3% cytotoxicity at a concentration of 20 mg/L at 4 and 6 h, respectively. 

## 3. Discussion

Ceragenins were developed to take advantage of the mechanisms of action of AMPs without the complications of relatively high production costs and instability in the presence of ubiquitous proteases. In this study, we observed that ceragenins displayed antibacterial activities at concentrations well below those required by cecropin, magainin, and LL-37 for the same activities. The AMPs have been well studied and compared to ceragenins; magainin, cecropin, and LL-37 were seen to be less effective against *P. aeruginosa* strains than ceragenins [15,16]. In addition, in parallel with the previously obtained data, ceragenins have been shown to have bactericidal activity [17]. The compound with the lowest MIC among ceragenins was determined as CSA-131 (4 mg/L). The structural differences of ceragenins explain the differences in their activities. Multiple series of ceragenins have been prepared to determine the structural features that combine to provide antibacterial activity. For example, the length of the lipid chain extending from the secondary amine in CSA-13 and in CSA-131 impacts MICs; CSA-13 contains an 8-carbon chain, while CSA-131 contains a 12-carbon chain. The additional hydrophobic carbons in CSA-131 help anchor the ceragenin on the bacterial membrane and result in lower MICs. Ceragenins containing ester groups (i.e., CSA-44, CSA-142, and CSA-144) were prepared to undergo partial bioresorption, similar to PLGA and other bioresorbable polymers. Replacement of the ether groups in CSA-13 with esters in CSA-44 results in a decrease in MICs.

According to cytotoxicity results against the A549 cell line, the cytotoxicities of AMPs are lower than those of the ceragenins. Among ceragenins, CSA-90, CSA-131, and CSA-138 showed the highest cytotoxicity, while LL-37 was determined as the highest cytotoxic AMP (IC_50_: 40.76 ± 5.35 mg/L) against the A549 cell line. However, The A549 cell line used in our study is a human lung cancer cell line, and therefore cytotoxicity with ceragenins and AMPs may be higher than with primary cell lines since AMPs and ceragenins show increased cytotoxic activity against transformed cells such as cancer cells and IB3-1 (cystic fibrosis lung cells) and are therefore considered as anticancer agents [18,19]. The membrane composition of cancer cells differs from primary (non-transformed) cells. It is known that the surface of cancer cells has negative charges that can associate with the positively charged structure of ceragenins and AMPs. Consequently, studies on transformed cancer cells may not provide fully accurate information regarding the cytotoxicity of ceragenins and AMPs. Studies involving primary cells may provide better measures of cytotoxicity. Nevertheless, these studies with the A549 cell line provide information about a minimum window of activity against bacteria without cytotoxicity to the eukaryotic cells. 

The fact that the cytotoxicity of ceragenins may limit their clinical use may be a significant problem. For this reason, studies are carried out to reduce the cytotoxicity of ceragenins through drug delivery systems [20,21]. The data we obtained in our study showed that encapsulation of CSA-131 with the pluronic drug delivery system significantly reduced cytotoxicity, but did not cause a significant change in its antimicrobial effect. While the MIC of CSA-131 was 4 mg/L, it was determined that the MIC of CSA-131P was 8 mg/L. Besides, MTT results showed that the IC_50_ of CSA-131 against A549 cells was determined as 11.37 ± 3.88 mg/L, while the IC_50_ value of CSA-131P was 40.05 ± 2.65 mg/L. However, when the effects of CSA-131 and CSA-131P on intracellular bacteria were compared, it was seen that CSA-131 was more effective. As can be seen in the MIC results, with extracellular bacteria, formulation with poloxamer did not significantly change the MIC. In this case, the CSA in the micelles has direct access to the bacteria. However, to access intracellular bacteria, the CSA has to get through the membrane of the eukaryotic cell. The decrease in cytotoxic activity suggests that there is less interaction with the membrane of the eukaryotic cell when the CSA is formulated in the poloxamer. So, it is likely that less CSA is getting into the eukaryotic cell when formulated in a poloxamer.

A study of intracellular bacteria showed that using gentamicin to treat an acute infection killed the extracellular bacteria; however, it did not prevent *P. aeruginosa* from remaining in the cell for at least three days [22]. For this reason, it is important to investigate the effects of antimicrobials not only on extracellular bacteria but also on intracellular bacteria. CSA-13 showed the highest intracellular activity in this study. At the concentration (10 mg/L) below the MIC (16 mg/L), CSA-13 reduced the intracellular bacterial count by about 3log. This may be due to the ceragenin-induced response of the cells, which kills pathogens. There is evidence that ceragenins trigger innate immune responses. LL-37 has also been shown to trigger LL-37 production [11]. In addition, Howell et al. showed that CSA-13 triggered the release of LL-37 [23]. Among the AMPs, LL-37 caused a one-log reduction at 20 mg/L. Noore et al. (2013) showed that LL-37 is effective not only against extracellular *S. aureus* but also against intracellular *S. aureus* [7]. In a study using osteoblast cell culture, the inhibitory effect of LL-37 on intracellular bacteria was found to be much higher than cefazolin and doxycycline. In a similar study, T24 (ATCC^®^ HTB4™), a human bladder epithelial cell line, was used and it was determined that the combination of LL-37 and ceragenins was more effective in reducing not only extracellular but also intracellular bacteria populations than their use alone [24]. 

Few ex-vivo or in-vivo biofilm studies with ceragenins and AMPs have been described. In a previous study, the efficacy of an LL-37-derived AMP in sinusitis induced by *P. aeruginosa* PO1 strain in rabbits was investigated. The study showed that high concentrations of LL-37 (2.5 g/L) are required for biofilm eradication. However, although high concentrations of an LL-37-derived peptide eradicated the biofilm and reduced the bacterial count, it exhibited pro-inflammatory and ciliotoxic effects on the sinus mucosa [25]. Similar to LL-37, CSA-131 has been shown impact cilia function at high concentrations (e.g., 100 mg/L). However, formulating CSA-131 in poloxamer micelles using Pluronic® F-127 eliminated cilia damage without changing the antimicrobial activity [20]. Similarly, in the present study, CSA-131P has lower cytotoxicity in the A549 cell line.

When the effects of antimicrobials on the adhesion of bacteria to the cell surface were examined, the most effective antimicrobial agent was determined as CSA-13. CSA-13 has been shown to inhibit bacterial adhesion at sub-MIC concentrations. However, it caused 40% cytotoxicity at a concentration of 5 mg/L at 6h in co-culture. According to our results, CSA-131P was more capable of inhibiting bacterial adhesion than CSA-131. Therefore, strategies such as drug delivery systems may need to be evaluated for the less cytotoxic, more potent CSA-13 molecule. 

## 4. Materials and Methods

### 4.1. Antimicrobial Agents and Bacterial Strain

Ceragenins (CSA-13, CSA-44, CSA-90, CSA-131, CSA-131-poloxamer (CSA-131P), CSA-138, CSA-142, CSA-144 and CSA-192) were synthesized from cholic acid. AMPs (magainin, cecropin, and LL-37) and Pluronic® F127 used to prepare CSA-131-poloxamer were obtained from Sigma-Aldrich (St. Louis, MO, USA). Stock solutions of ceragenins and AMPs were prepared by dissolving the materials in distilled water and storing at −20 °C for a maximum of one month. CSA-131 was combined with 5% Pluronic F-127 to form CSA-131-poloxamer and stored at room temperature. *P. aeruginosa*-GFP (ATCC^®^ 10145GFP™) containing a multicopy vector encoding the green fluorescent was used in the study.

### 4.2. Determination of MICs and MBCs

Studies for the determination of MICs and MBCs were conducted in accordance with the standards by CLSI [26,27]. MIC assays were performed using the broth microdilution method. After MICs were determined, 0.01 mL samples were taken from each well that did not show growth and plated TSA. MBCs were defined as the lowest antibiotic concentration providing at least 99.9% (three log) reduction in CFUs. All experiments were carried out in triplicate and mean values were calculated.

### 4.3. Cell Culture and Cytotoxixity

The human tumorigenic lung epithelial cell line A549 was purchased from the ATCC (CCL-185™). Cells were cultured in Dulbecco’s Modified Eagle Medium (DMEM) supplemented with 10% Fetal Bovine Serum (FBS), 2 mM L-glutamine and 1% penicillin-streptomycin solution and incubated at 37 °C in a humidified atmosphere with 5% CO_2_. 

For assays, 10^4^ cells were added to each well of a 96 flat-bottom microplate and kept in an incubator at 37 °C and 5% CO_2_ for 24 h. At the end of the incubation, increasing concentrations of antimicrobial agents (5, 10, 20, 50, 75 and 100 mg/L) were added to the wells and incubated for 24 h. The dye MTT was used to determine the cytotoxicities of the antimicrobial agents on the A549 cell line [18]. Experiments were repeated at least three times. The 50% inhibitory concentrations (IC_50_) were calculated by using GraphPad Prism version 8 (San Diego, CA, USA).

### 4.4. The Effects of Ceragenins and AMPs against Intracellular P. aeruginosa

The effects of 20, 10, 5, and 1 mg/L concentrations of ceragenins and AMPs against intracellular *P. aeruginosa* were determined. A population of 2 × 10^5^ cells was seeded into each well of a 24-well flat-bottom microplate. Cells were kept for 2 days until they became confluent. *P. aeruginosa*-GFP was added to the cells so that the infection rate (MOI: multiplicity of infection) was 2 (4 × 10^5^ CFU/mL). Plates were kept in an incubator with 5% CO_2_ at 37 °C for 1 h. After incubation, cells were washed twice with PBS. A solution of gentamicin (50 mg/L) was prepared, and 1 mL of solution was added to the wells and left in incubator for 1 h to eliminate extracellular bacteria [22]. After washing with PBS, antimicrobials were added to the cells and left to incubate overnight at 37 °C with 5% CO_2_. After washing with PBS, cells were lysed with 0.5% Triton-X solution in DMEM, and the number of intracellular viable bacteria released was determined. Aliquots of the resulting suspensions were removed from the wells and plated on TSA, after making the necessary dilutions. After incubation, colonies were counted. The experiments were repeated three times.

### 4.5. The Effects of Ceragenins and AMPs on P. aeruginosa Adhesion on A549 Cells

In a 24-well cell culture plate, 2 × 10^5^ cells were added to each well. Cells were kept in a 37 °C and 5% CO_2_ for one week to form a monolayer and the medium was changed at 2–3 days’ intervals [28,29,30]. In co-culture experiments, a different assay medium containing arginine was created to prevent disruption of the cell monolayer structure by bacteria. Bacterial populations were adjusted in the assay medium (DMEM, 10% FBS, 2 mM glutamine, and 0.4% arginine) and 375 µL of bacterial inoculum was added to wells to give a final population in the wells of 1 × 10^6^–6 × 10^6^ cfu/mL (MOI: 30:1). Aliquots of 125 µL of antimicrobials were added to give final concentrations of 20, 10, 5 and 1 mg/L were added with the bacteria. The co-culture obtained was incubated for 2, 4 and 6 h in an incubator with 5% CO_2_ at 37 °C. After incubation, the co-culture was washed 2–3 times with 0.5 mL PBS to remove planktonic bacteria and incubated for 30 min with 0.5% Triton-X in DMEM to disintegrate epithelial cells and biofilm. Samples were vortexed for 3 min, and the suspension was collected and serial dilutions were prepared. These dilutions were plated on TSA, incubated overnight at 37 °C, and colonies were counted the next day to determine cfu/mL [28,29,30]. The experiments were repeated three times.

### 4.6. Effect of Co-Culture on Cytotoxicity

The LDH cytotoxity test is a test based on the measurement of intracellular lactate dehydrogenase released into the culture medium as a result of cell death due to cell membrane damage. Since MTT also indicates bacterial death, a more specific test, the LDH test, was used to determine cell death. To determine the toxicity of bacteria and antibiotics on A549 cells, the supernatant with the biofilm was collected and stored at −20 °C before washing with PBS and destroying epithelial cells. This supernatant was centrifuged at 16,000× *g* in a centrifuge for 2 min. Aliquots of 10 µL of supernatant were evaluated twice according to the LDH cytotoxicity test kit (Promega, Madison, USA) protocol and LDH release was determined [31]. Results were read at 450 nm by a microplate reader (EON-BioTek Instruments, Winooski, VT, USA). Percent cytotoxicity was calculated. 

### 4.7. Fluorescence Microscopy

*P. aeruginosa*-GFP, producing green fluorescence protein, was used in the study. The density of biofilm formed in cells was determined using an Olympus BX51 fluorescence microscope and imaging with Olympus DP72 camera and DP2-TWAI software.

### 4.8. Statistical Analysis

GraphPad Prism program was used to statistically evaluate the results. Results were calculated as mean value ± standard deviation. Data analysis was performed using one-way analysis of variance (ANOVA) and Tukey analysis test as post hoc. *p* values lower than 0.05 were considered significant.

## 5. Conclusions

As a result, the potential of ceragenins and AMPs in the treatment of *P. aeruginosa* infections occurring in diseases such as CF has been demonstrated. According to the results of the study, ceragenins have advantages in terms of antimicrobial effect as well as structural and productive advantages over AMPs. But, the use of drug delivery systems may be a necessary strategy to avoid the cytotoxicity of ceragenins. 

## Figures and Tables

**Figure 1 pathogens-11-01044-f001:**
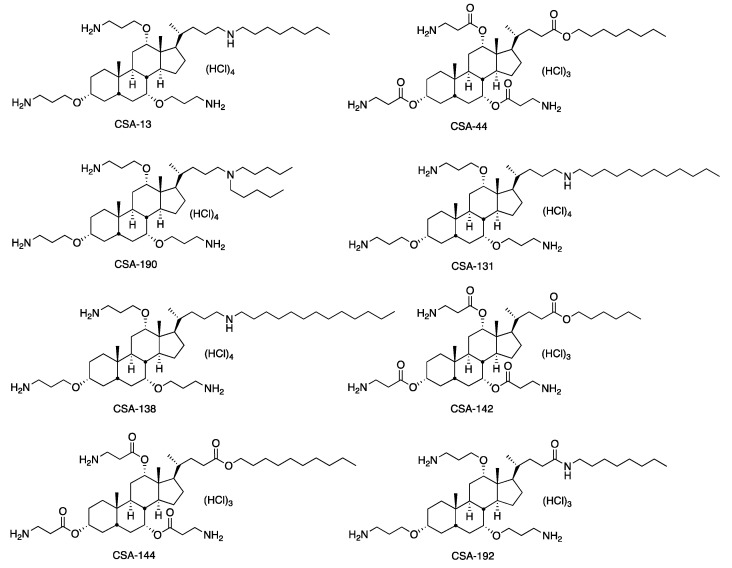
The chemical structures of ceragenins [12].

**Figure 2 pathogens-11-01044-f002:**
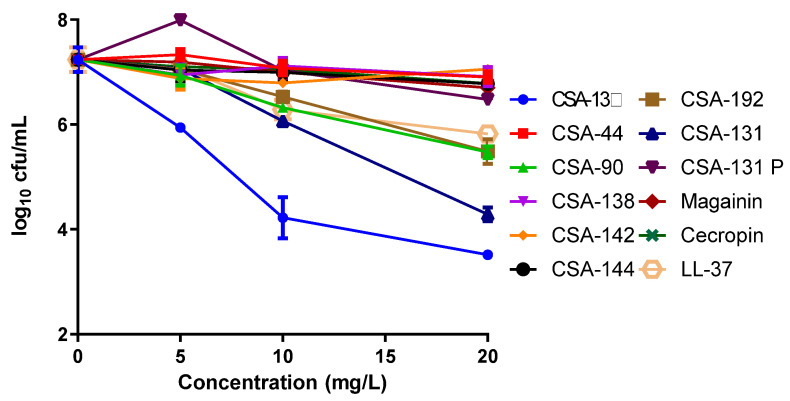
Effect of ceragenins and AMPs on intracellular *P. aeruginosa*.

**Figure 3 pathogens-11-01044-f003:**
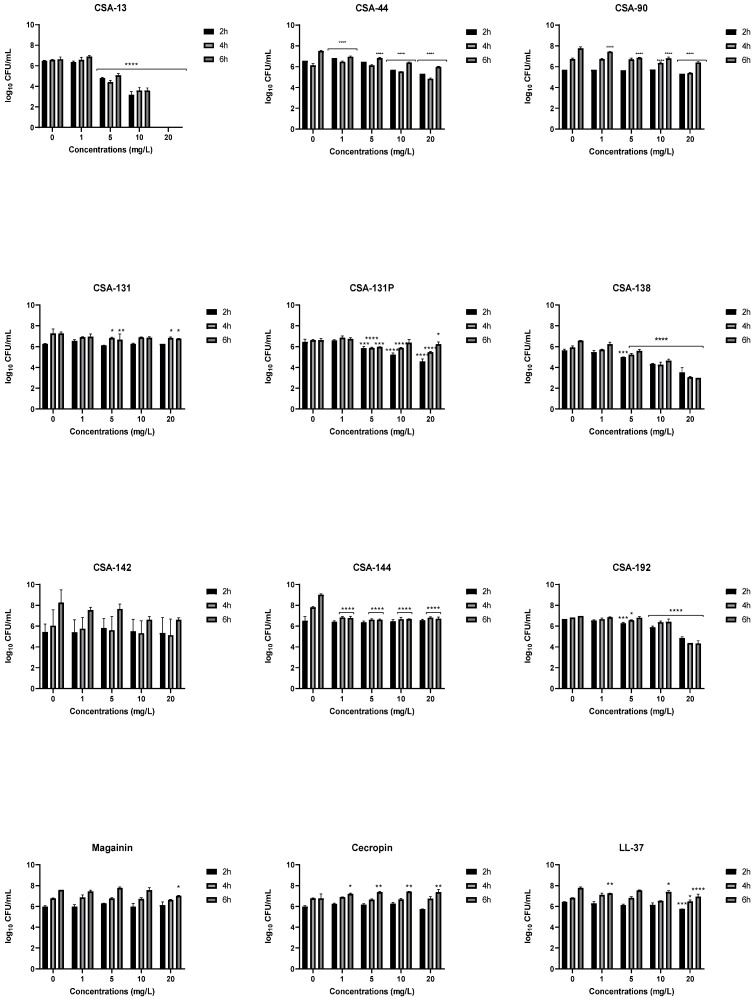
Inhibition of bacterial adhesion *, *p* < 0.05; **, *p* < 0.01; ***, *p* < 0.001; ****, *p* < 0.0001.

**Figure 4 pathogens-11-01044-f004:**
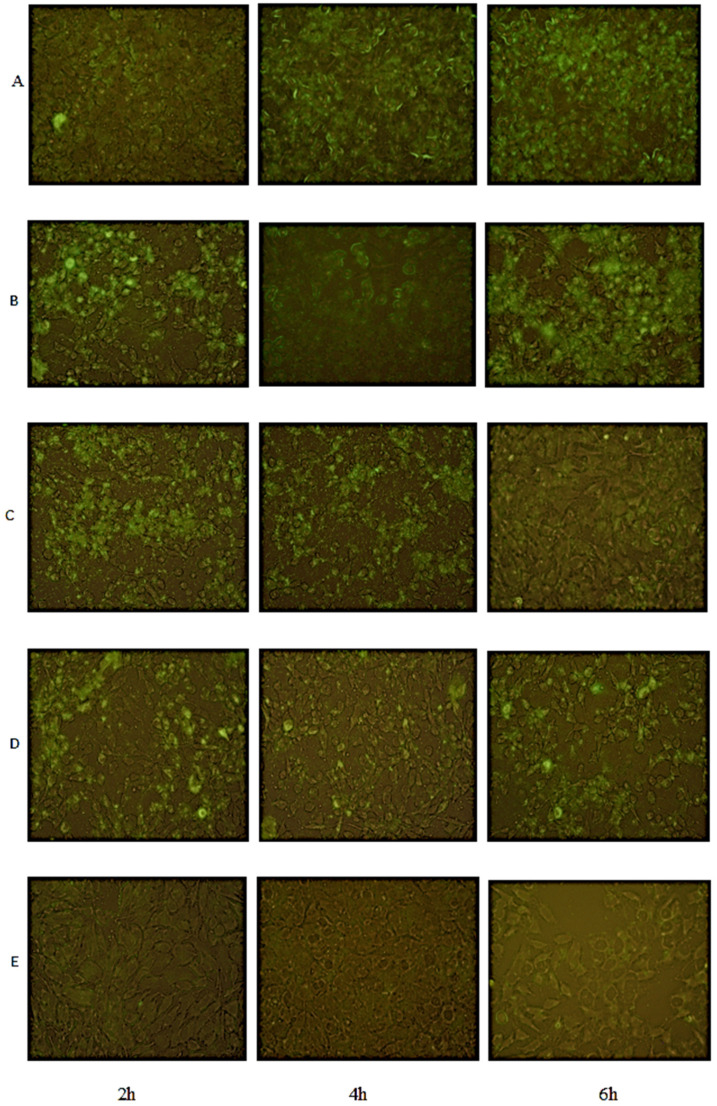
Fluorescence microscopy images of co-cultures of A549 cells with *P. aeruginosa*-GFP with and without treatment with CSA-13. From top to bottom (**A**): Control; (**B**): 1 mg/L; (**C**): 5 mg/L; (**D**): 10 mg/L; (**E**): 20 mg/L. Green color indicates *P. aeruginosa*-GFP (Magnification ×40, bar = 20 µm).

**Figure 5 pathogens-11-01044-f005:**
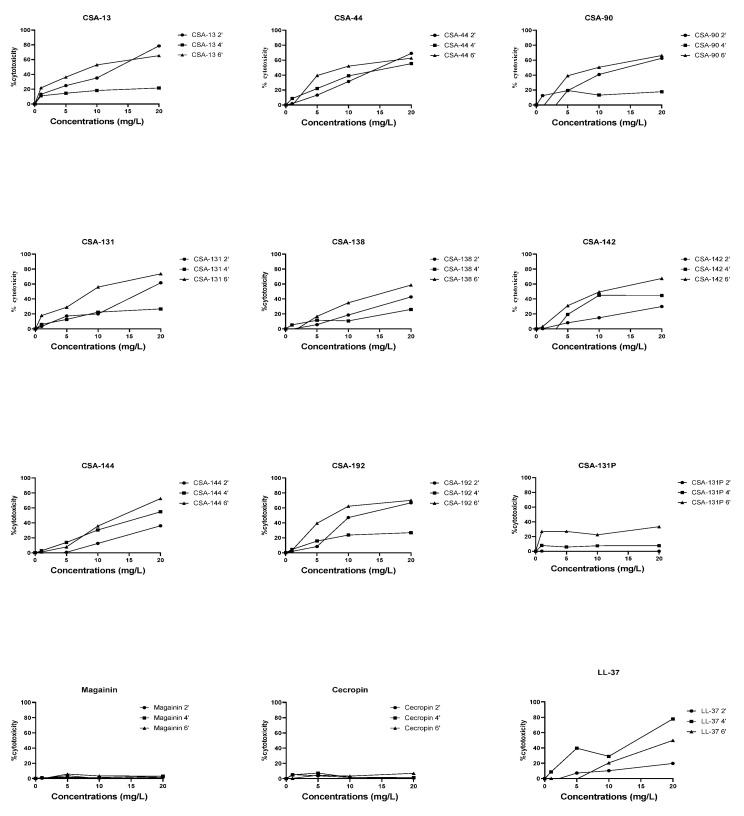
Cytotoxic effects of ceragenins and AMPs on co-culture as measured by release of LDH.

**Table 1 pathogens-11-01044-t001:** MIC and MBC values (mg/L) of ceragenins and AMPs.

	*P. aeruginosa*-GFP	
Antimicrobial Agents	MIC	MBC
CSA-13	16	32
CSA-44	8	16
CSA-90	32	64
CSA-131	4	4
CSA-131P	8	8
CSA-138	16	16
CSA-142	8	8
CSA-144	32	64
CSA-192	32	64
Magainin	>128	>128
Cecropin	>128	>128
LL-37	>128	>128

**Table 2 pathogens-11-01044-t002:** IC_50_ values of ceragenins and AMPs against A549 cells determined by MTT method.

Ceragenins	IC_50_ Values (mg/L)
CSA-13	20.68 ± 4.34
CSA-44	26.03 ± 0.95
CSA-90	11.62 ± 4.05
CSA-131	11.37 ± 3.88
CSA-131P	40.05 ± 2.65
CSA-138	11.78 ± 2.38
CSA-142	36.15 ± 1.25
CSA-144	26.03 ± 2.08
CSA-192	27.65 ± 5.02
Magainin	>200
Cecropin	>200
LL-37	40.76 ± 5.35

## Data Availability

Not applicable.

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
