# Peer review of "Effects of Ceragenins and Antimicrobial Peptides on the A549 Cell Line and an In Vitro Co-Culture Model of A549 Cells and Pseudomonas aeruginosa"

_pathogens, 2022, doi:10.3390/pathogens11091044_

Round 1
Reviewer 1 Report
The authors reported on the effects of ceragenins and antimicrobial peptides on an in vitro co-culture model of A549 cells and Pseudomonas aeruginosa. At least this is what the title says the text, however, is not so clear. Unfortunately, this manuscript is not well written and ill structured.
The intention of the study is not clear; the antimicrobial effects of ceragenins and AMPs have already been studied (Dosler, S.; Karaaslan, E. Inhibition and destruction of Pseudomonas aeruginosa biofilms by antibiotics and antimicrobial 356 peptides. Peptides 2014, 62, 32-37, doi:10.1016/j.peptides.2014.09.021. 357; Bucki, R.; Niemirowicz, K.; Wnorowska, U.; Byfield, F.J.; Piktel, E.; WÄ…tek, M.; Janmey, P.A.; Savage, P.B. Bactericidal 358 Activity of Ceragenin CSA-13 in Cell Culture and in an Animal Model of Peritoneal Infection. Antimicrob Agents Chemother 359 2015, 59, 6274-6282, doi:10.1128/aac.00653-15.) and a higher activity of ceragenins compared to AMPs against P. aeruginosa have been reported. So, why this study? What was the aim of this project? What is new and has not been reported before? The entire manuscript needs a clear concept, a working hypothesis and an explanation what is new.
Some explanations are needed what ceragenins and AMPs are and figure 5 should be referenced and placed in the introduction, not in the exp. part. Why have these two classes of compounds been chosen for this study but others ignored? Important information regarding the concept of the work appears in the discussion instead of the introduction or is missing at all.
A discussion section should discuss the results in the light of what was known already but this discussion is really confusing and does not clarify anything. The discussion seems to stress the effect of the studied compounds on cancer cells but the title is talking about biofilm formation. There seem also some results on the effect of carigenins and AMPs on established biofilms but this results have not been reported. What are the experimental details and where are the results on established biofilms discussed in the discussion?
A sentence explaining what LDH is and why it has been used for cytotoxicity tests would be highly appreciated. Why was for some studies the MTT but for others the LDH applied?
Line 82: in an MTT assay
For separation of decimals use dot not comma.
Figure 1: the symbols of CSA-13 and cecropin are almost the same, please use distinct ones.
Lines 119ff: This paragraph is not clear and needs rewriting. Why has cytotoxicity not been tested for A549 cells alone and compared to the cytotoxicity of A549 + P. aeruginosa? What has Pluronic F-127 to do with it? Where are the experimental results? It suddenly appears in the conclusion of this paragraph. Much better explanations are required here.
Line 159: from time to time? Do you mean in some stomach cancers? Please be more precise and explain better.
Maybe the authors can vary the use of the verb to determine a bit in the text.
Author Response
Dear Reviewer,
We are grateful to the reviewer for the valuable comments.
We have taken into account all suggestions and adjusted the manuscript accordingly.
In the following, we provide our point-by-point answers.
Comments and Suggestions for Authors
The authors reported on the effects of ceragenins and antimicrobial peptides on an in vitro co-culture model of A549 cells and Pseudomonas aeruginosa. At least this is what the title says the text, however, is not so clear. Unfortunately, this manuscript is not well written and ill structured.
- Dear reviewer, according to your comments, manuscript was rewritten to make it clear.
The intention of the study is not clear; the antimicrobial effects of ceragenins and AMPs have already been studied (Dosler, S.; Karaaslan, E. Inhibition and destruction of Pseudomonas aeruginosa biofilms by antibiotics and antimicrobial 356 peptides. Peptides 2014, 62, 32-37, doi:10.1016/j.peptides.2014.09.021. 357; Bucki, R.; Niemirowicz, K.; Wnorowska, U.; Byfield, F.J.; Piktel, E.; WÄ…tek, M.; Janmey, P.A.; Savage, P.B. Bactericidal 358 Activity of Ceragenin CSA-13 in Cell Culture and in an Animal Model of Peritoneal Infection. Antimicrob Agents Chemother 359 2015, 59, 6274-6282, doi:10.1128/aac.00653-15.) and a higher activity of ceragenins compared to AMPs against P. aeruginosa have been reported. So, why this study? What was the aim of this project? What is new and has not been reported before? The entire manuscript needs a clear concept, a working hypothesis and an explanation what is new.
- Ceragenins and antimicrobial peptides are two important classes of antimicrobials that are promising and therefore often used in researches. Although the antimicrobial effects of ceragenins and AMPs against Pseudomonas aeruginosa are known, those studies focused on antimicrobial activity. Cytotoxicity studies on these molecules are very limited. In this study, both the cytotoxicity of 12 antimicrobials and their simultaneous effects on both human cells and bacteria in cell-bacterial co-cultures were comprehensively examined and compared. In addition, the presence of intracellular bacteria is very important in terms of causing chronic infections and the effects of ceragenins on intracellular Pseudomonas aeruginosa were demonstrated for the first time in this study. Also, a pluronic F127-coated ceragenin to reduce cytotoxicity was also investigated. Introduction has been changed to better express the purpose of the study.
Some explanations are needed what ceragenins and AMPs are and figure 5 should be referenced and placed in the introduction, not in the exp. part. Why have these two classes of compounds been chosen for this study but others ignored? Important information regarding the concept of the work appears in the discussion instead of the introduction or is missing at all.
- More information was added about ceragenins and AMPs.
- Figure 5 was referenced and placed in the introduction.
- Although the discovery of new antimicrobials is extremely important, very few antimicrobial agents enter the market. In particular, the vast majority of antimicrobials entering clinical use are new members of existing antimicrobial classes or combinations. The presence of the new antimicrobial class, ceragenins and AMPs, is very important. Therefore, members of these two important classes were tested in our study.
- Some of the information in the discussion were moved in the introduction and the introduction was redesigned.
A discussion section should discuss the results in the light of what was known already but this discussion is really confusing and does not clarify anything. The discussion seems to stress the effect of the studied compounds on cancer cells but the title is talking about biofilm formation. There seem also some results on the effect of carigenins and AMPs on established biofilms but this results have not been reported. What are the experimental details and where are the results on established biofilms discussed in the discussion?
- The discussion section was changed according to the recommendations. Since we did the experiment up to 6 hours, according to the suggestion of the reviewers, “the biofilm formation” was changed to “adhesion of bacteria”. Results were discussed in the discussion.
A sentence explaining what LDH is and why it has been used for cytotoxicity tests would be highly appreciated. Why was for some studies the MTT but for others the LDH applied?
- “The LDH cytotoxity test is a test based on the measurement of intracellular lactate dehydrogenase released into the culture medium as a result of cell death due to cell membrane damage. Since MTT also indicates bacterial death, a more specific test, the LDH test, was used to determine cell death.” This explanation was added to the materyal and methods section (Effect of co-culture on cytotoxicity) to explain LDH and why it was used.
Line 82: in an MTT assay
- Corrected.
For separation of decimals use dot not comma.
- Corrections were made throughout the manuscript.
Figure 1: the symbols of CSA-13 and cecropin are almost the same, please use distinct ones.
- The symbols were changed. Also, the graph was changed as colorful in order to make it better look.
Lines 119ff: This paragraph is not clear and needs rewriting. Why has cytotoxicity not been tested for A549 cells alone and compared to the cytotoxicity of A549 + P. aeruginosa? What has Pluronic F-127 to do with it? Where are the experimental results? It suddenly appears in the conclusion of this paragraph. Much better explanations are required here.
- Thank you for your valuable comments. This paragraph was rewritten. The MTT test was performed at the 24th h only to obtain general information about the 24-h cytotoxicities of antimicrobials. However, cytotoxicities of antimicrobials on cell-bacterial co-cultures at 2nd, 4th and 6th h was evaluated by LDH test, since MTT test would also determine bacterial viability. The cytotoxic effect of the bacteria on A549 was also evaluated, but no significant effect was observed. Therefore, only the cytotoxicity of ceragenins was emphasized. In this paragraph, this situation was tried to be clarified.
Line 159: from time to time? Do you mean in some stomach cancers? Please be more precise and explain better.
- These sentence and paragraph were deleted.
Reviewer 2 Report
Oyardi and coworkers compare ceragenins and AMPs looking at their antimicrobial activity toward P. aeruginosa and their cytotoxic activity toward the pulmonary cell line A-549.
Unfortunately, the paper can not be presented as it is, and some points must be addressed.
Intro:
Authors should introduce and link better the topics, as well as provide more information E.g. line 52: what kind of mode of action do you refer to? I presume lytic, however AMPs have multiple mode of action, so more specificity is required. Line 63: Pluronics... this topic comes out of the blue, no explanation, no link with the other topics. This do not help the reader. Some info may be moved from discussion to intro.
Results:
Autrhors do not explain their results. Most often they just show the data. No rationale regarding why did they perform it, or no brief explanation of what could have been the expected output. This need to be done, avoiding however redundancies with mat/met and discussion session.
Moreover, there are some concerns:
Table 1. Ciprof. tested only on ATCC 27953, and atcc 27953 tested only woth ciprof This has no sense, as it does not provide any comparison. Should be deleted.
Figure 1: CSA13 cut the intracellular CFU number by about 3 logs... however, the tested concentration is lower even than the MIC of this compound. This looks quite strange. Author should explain or at least comment this point. Is it true? is it a compound-induced response of the cells that kills pathogens? this should be better elucidated.
Inhibition of biofilm fotmation: authors prolong the experiment up to 6 hours. Therefore it is more approrpiate to talk about bacterial adhesion to cells, or even uptake... but this range of time is quite short in order to talk of proper biofilm.
Figure 3: the quality is very poor. It gives just an idea of what is going on, but can not be used to provide strong evdences as it is.
MAt/met. #13 is superimposed to the structure of #44.
4.4 line 267: did the author check that gentamycin really removed all the planktonic and adherent bacteria? a control may be of great help.
Discussion:
Authors should evidence more the critical points of their compounds. I appreciated that they stated honestly that the therapeutic window is quite narrow, however this is just a sentence in a stream of words. The discussion may be modified in order to discuss a bit better the problems of ceragenins that the authors evidenced, maybe better discussing teh use of poloxamers (that by the way were never clearly presented in the paper)or other strategies. Moreover, also some hypothesis regarding structure/activity relationship to explain the different biological activity of the compounds may add some shent to the paper.
Author Response
Dear Reviewer,
We are grateful to the reviewer for the valuable comments.
We have taken into account all suggestions and adjusted the manuscript accordingly.
In the following, we provide our point-by-point answers.,
Authors should introduce and link better the topics, as well as provide more information E.g. line 52: what kind of mode of action do you refer to? I presume lytic, however AMPs have multiple mode of action, so more specificity is required. Line 63: Pluronics... this topic comes out of the blue, no explanation, no link with the other topics. This do not help the reader. Some info may be moved from discussion to intro.
- According to your comments, introduction was rewritten to link better the topics.
- Bactericidal mode of actions of AMPs and ceragenins are similar. So Line 52 was changed as “Ceragenins show lytic activity by destroying the bacterial cell membrane, similar to AMPs”.
- More information was given about drug delivery systems and pluronic.
Autrhors do not explain their results. Most often they just show the data. No rationale regarding why did they perform it, or no brief explanation of what could have been the expected output. This need to be done, avoiding however redundancies with mat/met and discussion session.
- All manuscript was reorganized in line with the suggestions.
Table 1. Ciprof. tested only on ATCC 27953, and atcc 27953 tested only woth ciprof This has no sense, as it does not provide any comparison. Should be deleted.
- Ciprofloxacin in Table 1 was deleted.
Figure 1: CSA13 cut the intracellular CFU number by about 3 logs... however, the tested concentration is lower even than the MIC of this compound. This looks quite strange. Author should explain or at least comment this point. Is it true? is it a compound-induced response of the cells that kills pathogens? this should be better elucidated.
- There is evidence that ceragenins (including CSA-13) trigger innate immune responses. The AMP LL-37 triggers production of LL-37. Howell, et al. showed that CSA-13 triggered release of LL-37. See the papers below. Explanation was added to the discussion.
Olekson, M. A.; Tao, Y.; Savage, P. B.; Leung, K. P. FEBS Open Bio 2017, 7, 953-967. Ceragenin peptide-mimics inhibit biofilms and affect mammalian cell bioability and migration in vitro.
Howell, M. D.; Streib, J. E.; Kim B. E.; Lesley, L.; Dunlap, A.; Geng, D.; Savage, P. B.; Leung, D. Y. M. J. Invest. Dermatol. 2009, 129, 2668-2675. Ceragenins: a new class of anti-viral compounds to treat orthopox infections.
Inhibition of biofilm fotmation: authors prolong the experiment up to 6 hours. Therefore, it is more approrpiate to talk about bacterial adhesion to cells, or even uptake... but this range of time is quite short in order to talk of proper biofilm.
- Thank you for highlighting this situation. As you said, it would not be correct to talk about mature biofilm in 6 hours. Biofilm formation changed as bacterial adhesion throughout the manuscript.
Figure 3: the quality is very poor. It gives just an idea of what is going on, but can not be used to provide strong evdences as it is.
- Focusing both cells and bacteria at the same time caused some quality loss. Some editing was made for figure 3 in order to better quality.
MAt/met. #13 is superimposed to the structure of #44.
- Structures of ceragenins placed in the introduction and corrected.
4.4 line 267: did the author check that gentamycin really removed all the planktonic and adherent bacteria? a control may be of great help.
- In this assay, this control was not performed. However, this issue will be taken into account in other studies. Nevertheless, according to a study conducted by Garcia-Medina et al. (2005), 30 min exposure of 50 μg/ml gentamicin killed extracellular P. aeruginosa PO1-GFP. This manuscript was given as a reference for this assay.
Garcia-Medina, R., Dunne, W. M., Singh, P. K., & Brody, S. L. (2005). Pseudomonas aeruginosa acquires biofilm-like properties within airway epithelial cells. Infection and immunity, 73(12), 8298-8305.
Authors should evidence more the critical points of their compounds. I appreciated that they stated honestly that the therapeutic window is quite narrow, however this is just a sentence in a stream of words. The discussion may be modified in order to discuss a bit better the problems of ceragenins that the authors evidenced, maybe better discussing teh use of poloxamers (that by the way were never clearly presented in the paper)or other strategies. Moreover, also some hypothesis regarding structure/activity relationship to explain the different biological activity of the compounds may add some shent to the paper.
- The discussion was modified. Especially, the importance of poloxamer was more discussed. Hypothesis regarding structure/activity relationship to explain the different biological activity of the compounds was also added to the paper.
Round 2
Reviewer 1 Report
The authors did a good job in revising the manuscript. They addressed all concerns.
Reviewer 2 Report
The authors addressed the major issues of the paper.